# INSTANTCHARACTER: PERSONALIZE ANY CHARACTERS WITH A SCALABLE DIFFUSION TRANSFORMER FRAMEWORK

## ABSTRACT

Current learning-based subject customization approaches, predominantly relying on U-Net architectures, suffer from limited generalization ability and compromised image quality. Meanwhile, optimization-based methods require subject-specific fine-tuning, which inevitably degrades textual controllability. To address these challenges, we propose InstantCharacter—a scalable framework for character customization built upon a foundation diffusion transformer. InstantCharacter demonstrates three fundamental advantages: first, it achieves open-domain personalization across diverse character appearances, poses, and styles while maintaining high-fidelity results. Second, we introduce a scalable dual-adapter architecture with stacked transformer encoders, which effectively processes open-domain character features and seamlessly interacts with the latent space of modern diffusion transformers. Third, to effectively train the framework, we construct a large-scale character dataset containing 10-million-level samples. The dataset is systematically organized into paired (multi-view character) and unpaired (text-image combinations) subsets. Our dual-adapter structure addresses the challenge of generating multi-character images by enhancing subject consistency through the image adapter and improving layout control of multiple subjects through the text adapter. Qualitative experiments demonstrate the advanced capabilities of InstantCharacter in generating high-fidelity, text-controllable, and character-consistent images, setting a new benchmark for character-driven image generation.

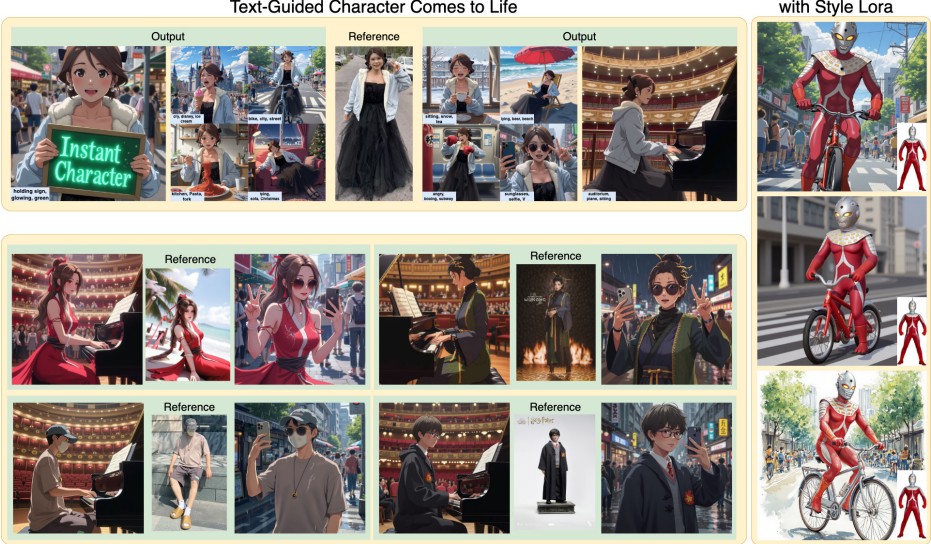

Figure 1: Open-domain character personalization with InstantCharacter.

# 1 INTRODUCTION

Character-driven image generation aims to create images that incorporates the user-defined character image and text prompts, playing a crucial role in various creative endeavors such as storytelling illustration, comic creation, game character design, and more. These capabilities enable a wide range of applications in entertainment, film production, e-commerce advertising, and beyond. Recent advancements in generative diffusion transformers have demonstrated unprecedented capabilities in synthesizing high-fidelity images from textual descriptions. Nevertheless, the potential of these state-of-the-art models for personalized image generation remains underexplored, especially in the context of creating character-driven visual narratives that embody human-like attributes.

Current methodologies for generating consistent images of specified subjects primarily rely on tuning- or adapter-based approaches. Adapter-based approaches Li et al. (2023); Ye et al. (2023); Mou et al. (2024) extract visual features through a subject encoder and integrate them into the image noise space via cross-attention mechanism. While these techniques achieve certain subject consistency and text controllability on UNet-based models, they struggle to personalize open-domain characters with diverse identities, poses, and styles. Although effective for customizing open-domain characters, tuning-based approaches Ruiz et al. (2023) require fine-tuning the model to reconstruct subject images, leading to long customization time and limited text controllability. Moreover, inference-time fine-tuning becomes computationally prohibitive for modern large-scale diffusion transformer models with billions of parameters.

Compared to traditional UNet-based architectures Rombach et al. (2022); Podell et al. (2023), modern Diffusion Transformers (DiTs) Esser et al. (2024); Labs (2024) exhibit powerful generative priors and offer unparalleled flexibility and capacity. However, fully unleashing their potential is non-trivial, as it requires a robust adapter network compatible with the framework to ensure alignment between character-specific features and vast generative latent space. In addition, training such an adapter necessitates adequate training data and effective training strategies. We observe that directly applying traditional adapters to large-scale DiTs often fails, as these adapters are primarily designed for UNet architectures and cannot scale effectively to models with billions of parameters, such as Flux Labs (2024) with 12 billion parameters.

To achieve generalized character personalization without compromising inference-time efficiency and textual editability, we propose InstantCharacter, a scalable diffusion transformer framework designed for character-driven image generation. InstantCharacter offers three key advantages: 1.**Generalizability.** It can flexibly personalize any character with different appearances, actions, and styles, ranging from photorealistic portraits to anime game assets. 2.**Scalability.** We develop a scalable dual-adapter architecture that can effectively integrate character features and interact with the latent space of modern DiTs. The image adapter interacts with noise tokens to inject character-specific features, while the text adapter engages with text tokens to enhance layout control by incorporating visual elements into textual embeddings. 3.**Versatility.** To enable efficient training, we collect a versatile 10-million-level character dataset, which contains paired (multi-view character) and unpaired (text-image combinations) subsets. Accordingly, we propose an efficient three-stage training strategy to accommodate heterogeneous data samples. Specifically, we decouple character consistency (unpaired data), textual controllability (paired data), and image fidelity (high-resolution data) to prevent mutual interference between high-fidelity identity maintenance and prompt-guided character manipulations.

We implement InstantCharacter based on the powerful FLUX1.0-dev model. Qualitative and quantitative comparisons with previous work demonstrate InstantCharacter's advanced capabilities in generating high-fidelity, text-controllable, and character-consistent images.

# 2 RELATED WORK

**T2I Diffusion Models.** Recent advances Esser et al. (2024); Podell et al. (2023); Rombach et al. (2022) in text-to-image generation have witnessed a paradigm shift from traditional U-Net architectures Rombach et al. (2022) to more powerful diffusion transformers Esser et al. (2024) (DiTs). While early diffusion models such as stable diffusion (SD) demonstrated remarkable image synthesis capabilities, modern DiT-based systems like SD3 Esser et al. (2024) and FLUX.1 Labs (2024) have set new benchmarks in generation quality through their transformer-based architectures and advanced

techniques like rectified flows. This architectural evolution presents both opportunities and challenges for character-centric applications, while DiTs offer superior generation capacity, their adaptation for identity-preserving tasks remains largely underexplored. Our work bridges this critical gap by developing the first DiT-based framework specifically optimized for character customization.

**Personalized Character Generation.** Recent advances in personalized image generation have evolved from tuning-based to adapter-based approaches. Early methods Ruiz et al. (2023); Chefer et al. (2023); Feng et al. (2025); Kumari et al. (2023); Gal et al. (2022) relied on fine-tuning the entire diffusion model for each new subject, which was computationally expensive and suffered from poor generalization due to limited training data. To address these issues, recent works Ye et al. (2023); Li et al. (2023); Mou et al. (2024); Wang et al. (2024b); Li et al. (2024); Huang et al. (2024); Wang et al. (2024a); Mao et al. (2024); Song et al. (2024); Gal et al. (2023) introduced adapter-based techniques that avoid test-time fine-tuning. For instance, IP-Adapter Ye et al. (2023) employs a clip image encoder to extract subject features and injects them into a frozen diffusion model via cross-attention, enabling efficient personalization. However, these adapter-based methods are built upon UNet-based architectures with restricted capacity, causing them to struggle in low-fidelity outputs and limited generalization ability.

While some concurrent works Tan et al. (2024); Zhang et al. (2025); Mao et al. (2025) have also utilized DiT-based models for image customization, they typically concatenate the condition image tokens and noise tokens, and train LoRAs to endow DiT models with customization capabilities. Our approach is distinguished by the introduction of a novel dual-adapter architecture that does not require training LoRAs, which are known to often compromise the inherent image fidelity of the base model. Furthermore, our adapter design demonstrates superior capability in modeling subject-specific details, thereby enabling our method to achieve highly consistent character customization.

## 3 METHODOLOGY

Modern DiTs Esser et al. (2024); Labs (2024) have demonstrated unprecedented fidelity and capacity compared to traditional UNet-based architectures, offering a more robust foundation for generation and editing tasks. Building upon these advances, we present InstantCharacter, a novel framework that extends DiT for generalizable and high-fidelity character-driven image generation. As illustrated in Fig. 2 and Fig. 3, InstantCharacter's architecture centers around three key innovations. First, a scalable dual-adapter architecture is developed to effectively parse character features and seamlessly interact with DiTs latent space. Second, a progressive three-stage training strategy is designed to adapt to our collected versatile dataset, enabling separate training for character consistency and text editability. Third, we devise a new pipeline for constructing training data pairs for multi-character customization. By synergistically combining flexible adapter design and phased learning strategy, we enhance the general character customization capability while maximizing the preservation of the generative priors of the base DiT model. In the following sections, we will detail the adapter's architecture and elaborate on our progressive training strategy.

### 3.1 SCALABLE DUAL-ADAPTER DESIGN

Recent DiT models Esser et al. (2024); Labs (2024) employ a multi-modal attention mechanism that enables deep interaction and fusion between image and text modalities. To accommodate this architecture, we propose a dual-adapter architecture to integrate information from reference images into both image tokens and text tokens separately. To better adapt to DiT models, we propose a full-transformer structure that enables scalability by increasing layer depth and hidden feature sizes. As shown in Fig. 2, our approach features two distinct adapters: an image adapter and a text adapter. The image adapter is designed to interact with noisy image tokens, allowing for the effective injection of character-specific features into the image generation process. Meanwhile, the text adapter engages with text tokens to enhance layout control and achieve better separation of multiple characters by integrating visual features into their respective textual embeddings. Subsequently, we will present a detailed introduction to the two adapter modules.

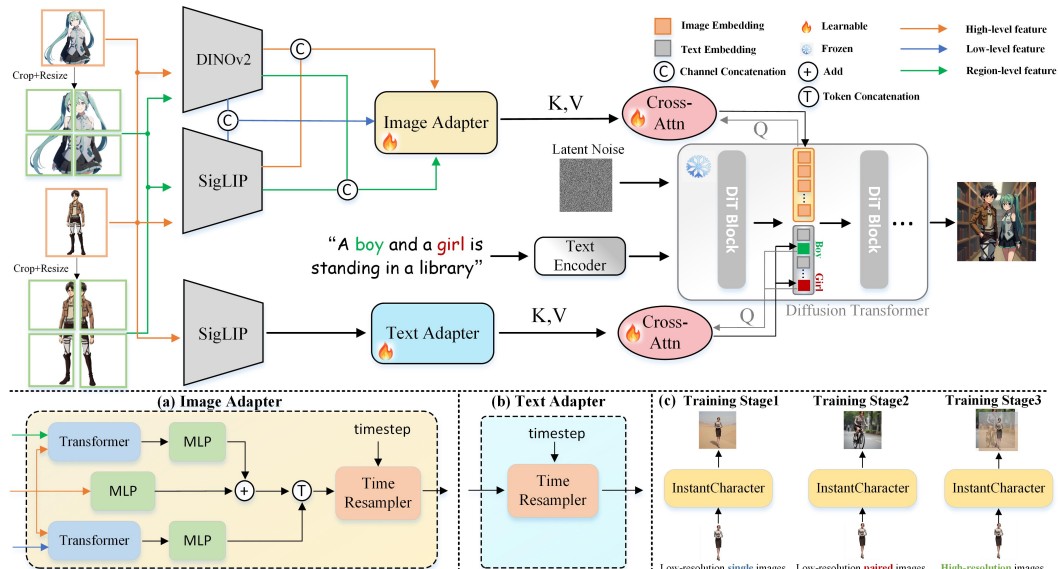

Figure 2: Our framework seamlessly integrates a scalable dual-adapter architecture with a pretrained DiT model. The image adapter consists of multiple stacked transformer encoders that incrementally refine character representations, enabling effective interaction with the latent space of the DiT. The text adapter is a time resampler that integrates visual features into text embeddings. The training process employs a three-stage progressive strategy, beginning with unpaired low-resolution pretraining and culminating in paired high-resolution fine-tuning.

### 3.1.1 IMAGE ADAPTER

This adapter is designed to inject abundant character features into the noise space, thereby improving character consistency of the generated images. We first leverage pre-trained large vision foundation encoders to extract general character features, benefiting from their open-domain recognition abilities. Previous methods Ye et al. (2023); Li et al. (2024) typically rely on CLIP Radford et al. (2021) for its aligned visual and textual features. However, while CLIP is capable of capturing abstract semantic information, it tends to lose detailed texture information, which is crucial for maintaining character consistency. To this end, we replace CLIP with SigLIP Zhai et al. (2023), which excels in capturing finer-grained character information. In addition, we introduce DINOv2 Oquab et al. (2023) as another image encoder to enhance the robustness of features, reducing the loss of features caused by background or other interfering factors. Finally, we integrate DINOv2 and SigLIP features via channel-wise concatenation, resulting in a more comprehensive representation of characters.

**Transformer Encoders:** Since SigLIP and DINOv2 are pre-trained and inferred at a relatively low resolution of 384, the raw output of general vision encoders, denoted by $F^{\text{siglip}} \in R^{n \times c1}$ and $F^{\text{dino}} \in R^{n \times c2}$ where $n$ and $c1, c2$ denote the number of tokens and channels, may lose fine-grained features when processing high-resolution character images. To mitigate this issue, we employ a dual-stream feature fusion strategy to explore **low-level** and **region-level** features, respectively. First, we directly extract low-level features, denoted by $F_l^{\text{siglip}} \in R^{n \times c1}$ and $F_l^{\text{dino}} \in R^{n \times c2}$, from the shallow layers of the general vision encoders, capturing details that are often lost in higher layers. Second, we divide the reference image into multiple non-overlapping patches and feed each patch into the vision encoder to obtain region-level features, denoted by $F_r^{\text{siglip}} \in R^{n \times c1}$ and $F_r^{\text{dino}} \in R^{n \times c2}$. Then these two distinct feature streams undergo hierarchical integration through dedicated intermediate transformer encoders, as shown in Fig. 2 (a). Specifically, each feature pathway is independently processed by a separate transformer encoder to integrate with high-level semantic features, which can be formulated as:

$$A = \text{Attention}(F^Q, F_r^K, F_r^V), \quad \text{and} \quad A = \text{Attention}(F^Q, F_l^K, F_l^V), \tag{1}$$

where Attention denotes the standard transformer attention operation. $F \in R^{n \times (c1+c2)}$ denotes the concatenated feature of $F^{\text{siglip}}$ and $F^{\text{dino}}$ in channel dimension. The same concatenation strategy is

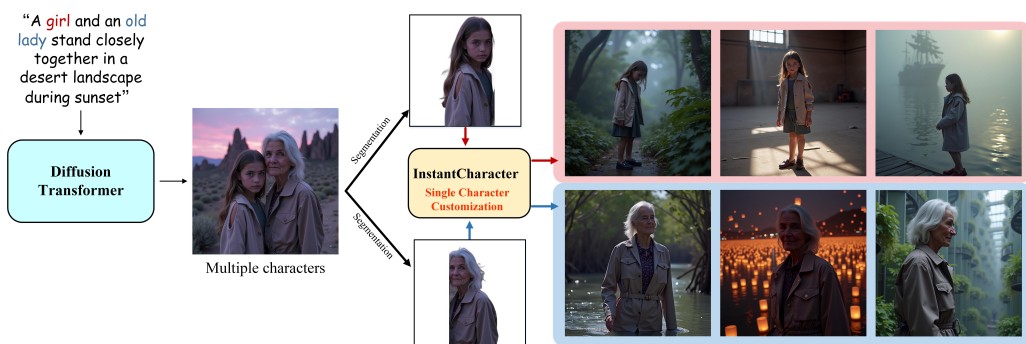

Figure 3: Multi-character data construction pipeline.

applied to $F_l$ and $F_r$. Subsequently, the refined feature embeddings from both pathways are concatenated along the token dimension, denoted by $F_{\text{concat}} \in R^{2n \times (c1+c2)}$, establishing a comprehensive fused representation that captures multi-level complementary information.

**Time Resampler:** The refined character features $F_{\text{concat}}$ are projected into the denoising space via a projection head and interact with the latent noise. We implement this through a timestep-aware Q-former Ye et al. (2023) that processes $F_{\text{concat}}$ as key-value pairs while dynamically updating a set of learnable queries $F_I$ through attention mechanisms, i.e., $F_I = \text{Attention}(F_I^Q, F_{\text{concat}}^K, F_{\text{concat}}^V)$. The transformed query features $F_I$ are then injected into the denoising space via learnable cross-attention layers. Denote the image hidden states of the DiT blocks as $H_I$, the cross-attention can be formulated as follows:

$$H_I = H_I + \text{Attention}(H_I^Q, F_I^K, F_I^V), \tag{2}$$

For the multi-character scenario, we employ several routing masks following existing works He et al. (2025; 2024), and the cross attention can be simply modified as follows:

$$H_I = H_I + \sum_{i=1}^{m} M_i \cdot \text{Attention}(H_I^Q, F_{I_i}^K, F_{I_i}^V), \tag{3}$$

where $m$ denotes the number of characters and $M_i$ denotes the corresponding character mask.

### 3.1.2 TEXT ADAPTER

Although the image adapter facilitates the interaction between character features and the latent noise space, it struggles to separate multiple characters, often leading to the mixing of different character features. Given that the text prompt can control the overall layout of the generation and effectively distinguish different characters, we introduce an additional text adapter. This adapter injects the features of each subject into the corresponding text tokens, allowing for precise control and differentiation of individual characters within the generated content. Specifically, the text adapter integrates character image features $F_i^{siglip}$ where $i \in \{1, \cdots, m\}$ into a group of learnable tokens $F_{Ti}$ through a timestep-aware Q-Former Ye et al. (2023). We denote the text tokens for the $i$-th character as $C_{Ti}$. Consequently, these unique character features are embedded within the text tokens through a cross-attention mechanism:

$$C_{Ti} = C_{Ti} + \text{Attention}(C_{Ti}^Q, F_{Ti}^K, F_{Ti}^V), \quad \text{where } i \in \{1, \cdots, m\}. \tag{4}$$

### 3.2 TRAINING STRATEGIES

To enable effective training of the framework, we first curate a high-quality dataset of 10 million images containing diverse full-body humans/characters, including both unpaired images for learning robust character consistency and paired sets for achieving precise text-to-image alignment.

As shown in Fig. 2 (c), our training regimen is meticulously designed to optimize character consistency, text controllability, and visual fidelity. To achieve character consistency, we first train with unpaired data, where the character image is incorporated as reference guidance to reconstruct itself and preserve

structural consistency. We discovered that using a resolution of 512 is significantly more efficient than 1024.

In the second phase, we continue training at a low resolution (512) but switch to paired training data. By taking the character image as input, we aim to generate images of the character in different actions, poses, and styles within a new scene based on a given textual description. This training stage efficiently eliminates the copy-paste effect and enhances text controllability, ensuring that the generated images accurately follow the textual condition.

The final phase involves high-resolution joint training using both paired and non-paired images. We found that a limited number of high-resolution training iterations can substantially improve the visual quality and texture of the images. This stage leverages high-quality images to achieve high-fidelity and textually controlled character images.

## 3.3 MULTI-CHARACTER DATA CONSTRUCTION PIPELINE

Obtaining paired multi-character images is challenging, yet crucial for enhancing multi-character personalization performance. Traditional approaches Xiao et al. (2024a); He et al. (2025) use single images containing multiple characters as the training dataset and reconstruct the image using the character features from the image itself. However, this method can easily result in copy-paste effects, as seen in the fourth column of Fig. 6, similar to what occurs in our training stage 1. By leveraging the exceptional performance of the base model in synthesizing multi-subject images and the outstanding capabilities of InstantCharacter in generating single-subject customizations, we can easily construct paired multi-character images and reuse them to train InstantCharacter.

We thus propose a data construction pipeline illustrated in Fig. 3: (1) we first employ a DiT-based image generation model to create a high-quality image featuring multiple characters; (2) we then use an existing image segmentation method Zheng et al. (2024) to separate the two characters from the generated image; and (3) finally, using each segmented character as a reference image, we generate single-character images with diverse backgrounds, actions, and viewpoints by feeding the model with varied prompts.

## 4 EXPERIMENTS

### 4.1 EXPERIMENTAL SETUP

**Implementation Details:** We utilize Flux as our pretrained diffusion transformer model. The image adapter is designed as two four-layer transformers, with input queries derived from high-level semantic features, and key and value features specifically crafted from low-level and region-level features. The time resampler (for both image and text adapters) adopts a similar approach to IPAdapter Ye et al. (2023). Our model undergoes training in three stages with batch sizes of 192, 192, and 64, for 200K, 50K, and 50K iterations, respectively, across 64 NVIDIA H20 GPUs. We employ a learning rate of $2e - 5$ with a warm-up phase of 2000 steps.

**Datasets:** In addition to the off-the-shelf benchmarks of OmniContext Wu et al. (2025a) and Unsplash-50 Gal et al. (2024), we constructed a new benchmark named Character350, comprising 350 test cases. This dataset was created by manually curating 35 canonical subjects from an extensive collection of characters and designing 10 distinct prompts for each. To evaluate the performance on more than a single character, we select 105 samples containing two characters from Character350 as the Two-Character105 benchmark. We report the performance on Character350 and Two-Character105 in main paper and present more results on other benchmarks in the supplementary files.

**Objective Metrics:** Following existing work Ruiz et al. (2023), we evaluate our model using DINO-I Caron et al. (2021), CLIP-I, CLIP-T, and Image Reward (IR) Xu et al. (2023) scores. DINO-I and CLIP-I are employed to assess subject similarity. To minimize background interference, we calculate subject similarity after segmenting both the reference and generated subjects using Language SAM Kirillov et al. (2023). For evaluating text controllability, we use CLIP-T and Image Reward (IR). The CLIP-T metric is determined by calculating the cosine similarity in the CLIP text-image embedding space. Additionally, ImageReward, which is considered a more reliable metric that aligns with human preferences, is utilized to further assess controllability.

**Subjective Metrics:** Besides, we provide subjective metrics to provide comprehensive studies. Specifically, we first introduce Gemini-T and Gemini-I by repurposing Gemini to evaluate the text controllability and subject consistency scores. Please refer to the supplementary file for more details. Furthermore, we conducted a user study where 50 participants subjectively assessed the generated images from different methods, focusing particularly on identity preservation and fine detail fidelity.

**Baselines:** For single-character generation, we primarily compare our method against DiT-based approaches, which currently achieve state-of-the-art performance. Our baselines include FLUX-based methods: OminiControl Tan et al. (2024), EasyControl Zhang et al. (2025), ACE++ Mao et al. (2025), and UNO Wu et al. (2025b). For multi-subject generation, we compare to MS-Diffusion Wang et al. (2024c), OmniGen Xiao et al. (2024b), and UNO Wu et al. (2025b). For evaluation, we curated open-domain character images excluded from training; images and prompts are provided in the supplementary material.

Table 1: Quantitative results on Character350 benchmark.

|  | Objective Metrics | | | | Subjective Metrics | | |
|---|---|---|---|---|---|---|---|
|  | IR↑ | CLIP-T↑ | CLIP-I↑ | DINO↑ | Gemini-I↑ | Gemini-T↑ | User study (Win Rate)↓ |
| **Ours** | 0.994 | 0.308 | 0.795 | 0.604 | 8.300 | 9.137 | - |
| **OminiControl** | 0.564 | **0.312** | 0.697 | 0.490 | 3.480 | 7.500 | 85% |
| **ACE++** | 0.707 | 0.297 | 0.763 | 0.610 | 5.171 | 8.457 | 80% |
| **UNO** | 0.051 | 0.284 | **0.849** | **0.765** | **8.720** | 7.154 | 80% |
| **EasyControl** | **1.230** | **0.312** | 0.678 | 0.495 | 2.351 | **9.691** | 67% |
| **DSD** | 0.954 | 0.304 | 0.696 | 0.541 | 3.354 | 8.849 | 74% |
| **OneDiffusion** | 0.889 | 0.301 | 0.699 | 0.506 | 4.300 | 9.023 | 76% |

Table 2: Quantitative results on Two-Character105 dataset

|  | IR↑ | CLIP-T↑ | Gemini-T↑ | CLIP-I↑ | DINO↑ | Gemini-I↑ |
|---|---|---|---|---|---|---|
| Ours | 0.491 | 0.301 | **7.895** | 0.641 | 0.509 | **9.210** |
| UNO | 0.593 | 0.302 | 7.105 | 0.735 | 0.642 | 6.210 |
| OmniGen | **0.625** | **0.304** | 6.600 | **0.808** | 0.722 | 7.714 |
| MS-Diffusion | 0.255 | 0.303 | 5.486 | 0.793 | **0.724** | 8.314 |
| OneDiffusion | 0.491 | 0.301 | 7.543 | 0.641 | 0.509 | 1.019 |

## 4.2 COMPARISON WITH BASELINES

**Qualitative Results:** Our analysis of **single-character customization**, as illustrated in Fig. 4, reveals distinct limitations in existing methods. While OminiControl and EasyControl struggle to preserve character identity features, ACE++ maintains only partial fidelity in simple scenarios but falters with action-oriented prompts. UNO, on the other hand, enforces excessive consistency at the cost of text editability, particularly for generating actions and backgrounds. In contrast, InstantCharacter consistently outperforms these approaches, achieving superior preservation of character details with high fidelity and precise text controllability. This advantage stems from our proposed image adapter, which excels at capturing open-domain character details through region-level and low-level transformer encoding, as further supported by the quantitative measurements in Tab. 1.

As illustrated in Fig. 5, our approach also excels in **multi-subject customization**, with InstantCharacter maintaining robust character consistency and precise text alignment. In contrast, UNO frequently loses essential character details, while OmniGen and MS-Diffusion exhibit failures in feature preservation or introduce artifacts (e.g., erroneous additions/omissions). InstantCharacter's multi-subject capability stems from strong single-subject performance, enhanced by a dedicated text adapter for multi-subject generation. Additional qualitative results are included in the supplementary material due to space limitations.

**Quantitative Results:** We evaluate subject consistency and text editability across methods, with results summarized in Tab. 1 and Tab. ftab:twocharacter105. While UNO achieves the highest CLIP-I/DINO-I scores for subject consistency, it comes at the cost of poor text controllability. Fig. 4 reveals that its generations often exhibit low-quality customizations with incoherent backgrounds and actions, reflected in its subpar CLIP-T and ImageReward metrics. Conversely, OminiControl and ACE++ perform well on CLIP-T/ImageReward but underperform on other consistency measures. Our approach strikes an optimal balance: it maintains competitive consistency scores while achieving superior text alignment, as evidenced by both quantitative metrics and qualitative results. Note that in the user study, our method outperforms previous methods significantly, we achieve overall higher win rates, which further validates the superiority of our approach.

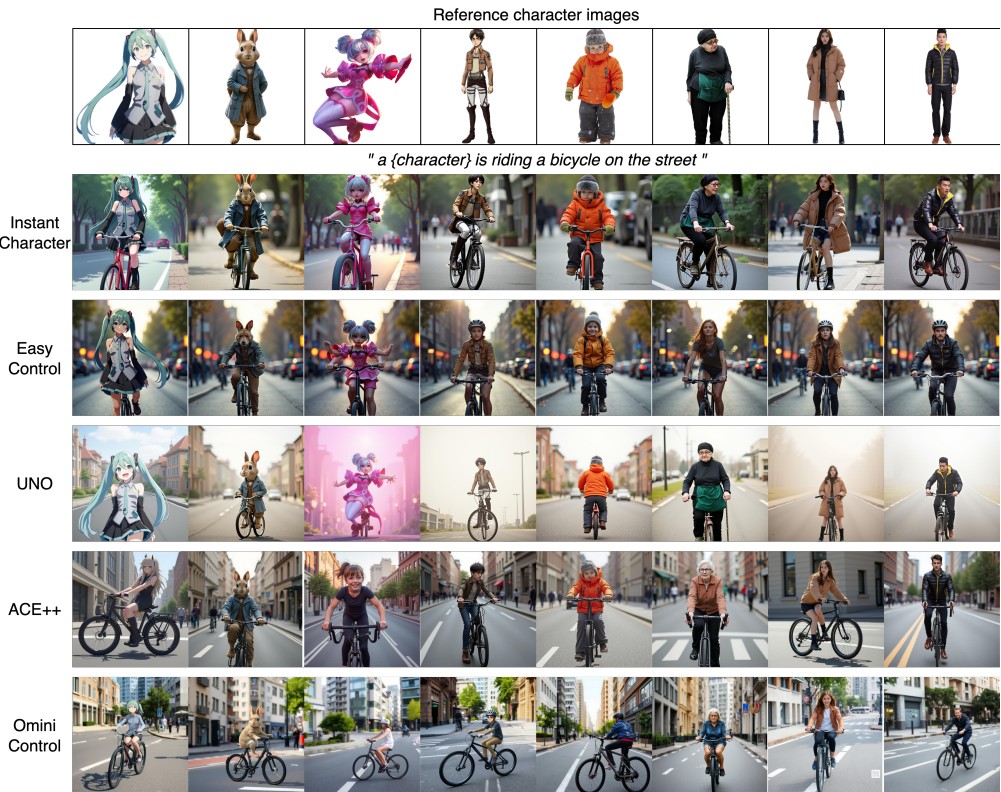

Figure 4: Qualitative comparison on character personalization. Our method generally shows the best image fidelity and character consistency while maintaining the desirable textual controllability.

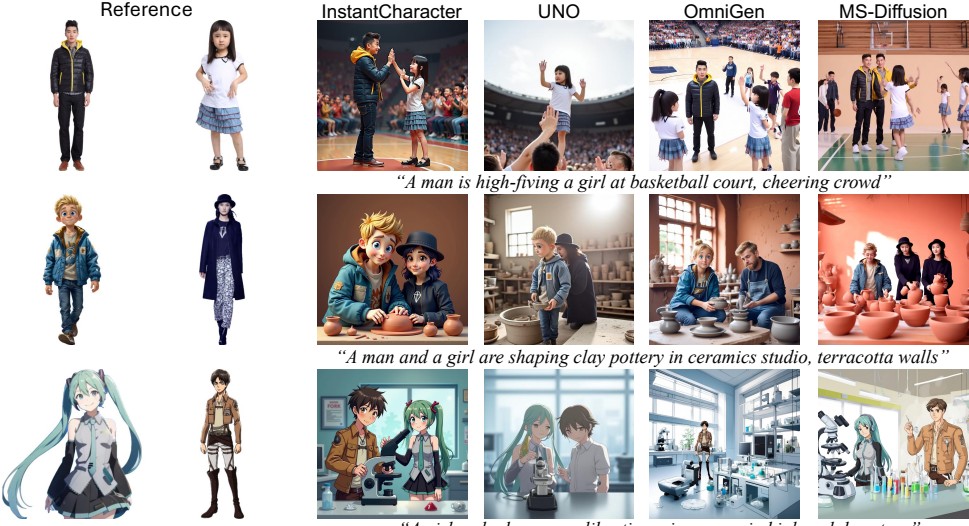

Figure 5: Qualitative comparison on multi-character personalization.

**Ablation Study:** We investigate the comprehensive ablation study to investigate the effects of different components. As shown in th Tab. 3, since Transformers Encoder (TE) is designed to integrate both fine-grained low-level and region-level features. The absence of TE leads to a significant performance drop with -8.9% CLIP-I and -8.3% DINO scores, underscoring its crucial role in identity preservation. A similar performance degradation is observed upon removing the time-resampler (TR) module. This is because the TR module serves as a bridge between reference image features and the noise space, enabling more effective integration of character features to enhance consistency, while also allowing for flexible adaptation to complex text-driven modifications. By integrating all proposed modules,

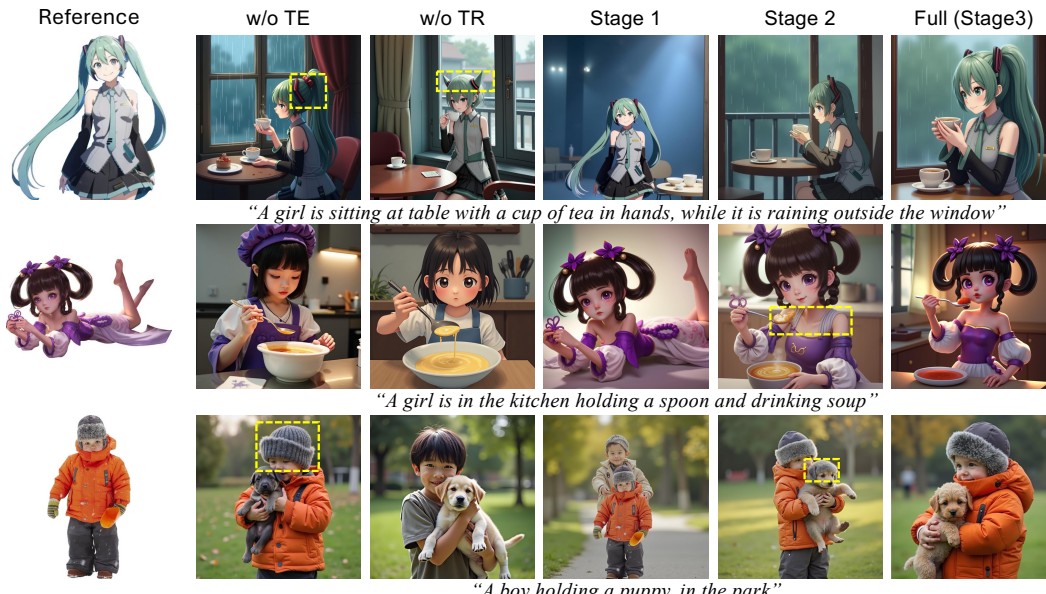

Figure 6: Qualitative ablation results.

Table 3: Ablation on Character350 bechmark.

| | IR↑ | CLIP-T↑ | Gemini-T↑ | CLIP-I↑ | DINO↑ | Gemini-I↑ |
|---|---|---|---|---|---|---|
| Ours-Full | 0.994 | 0.308 | 9.137 | 0.795 | 0.604 | 8.300 |
| Ours-Stage2 | 0.788 | 0.304 | 8.714 | 0.793 | 0.611 | 7.957 |
| Ours-Stage1 | -0.870 | 0.258 | 4.631 | **0.928** | **0.877** | **9.586** |
| Ours w/o TE | 0.935 | 0.312 | 9.163 | 0.706 | 0.521 | 3.997 |
| Ours w/o TR | **1.313** | **0.318** | **9.769** | 0.612 | 0.399 | 0.606 |

our model achieve best balance between visual quality and textual faithfulness. We also show some qualitative comparisons in Fig. 6. We can observe that omitting either the TR or TE module leads to a significant decrease in the character consistency. This demonstrates that both modules play a crucial role in accurately extracting character features.

We further analyze the model's performance across training stages. Although the model achieves the best CLIP-I and DINO scores after Stage 1 (trained on low resolution unpaired images), it suffers from copy-paste behavior, achieving high character consistency but poor text controllability, as shown in the 4-th column in Fig. 6. Stage 2 (trained on low resolution paired images) substantially improves text alignment. Finally, Stage 3 (trained high resolution images) enhances visual quality while preserving both consistency and controllability. Fig. 6 qualitatively validates the necessity of this multi-stage training paradigm.

## 5    CONCLUSION

We present InstantCharacter, an innovative diffusion transformer framework that significantly advances character-driven image generation. Our solution delivers three fundamental advantages: first, it achieves unprecedented open-domain personalization across diverse character appearances, poses, and styles while preserving high-fidelity quality; second, it develops a scalable dual-adapter architecture that effectively processes character features and interacts with diffusion transformers' latent space; third, it establishes an effective three-stage training methodology to separately optimize character consistency and textual control. Qualitative results validate InstantCharacter's superior performance in generating high-fidelity, character-consistent, and text-controllable images. More broadly, our work offers insights for adapting foundation diffusion transformers to specialized generation tasks, potentially inspiring new developments in controllable visual synthesis.

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

# A  APPENDIX

In this appendix, we begin by claiming the LLM usage in the paper and the ethics statement, and we then detail the training objectives for both single and multiple character customization models. Next, we offer a clear visualization of the training data, which includes unpaired single images as well as paired images featuring single and multiple characters. Following this, we provide additional experimental results including more numerical performance on extended benchmarks and more visualization results across single and multiple character customizations. Finally, we discuss the limitations of InstantCharacter and propose potential improvements.

**LLM Usage:** We would like to disclose that during the preparation of this manuscript, we utilized a large language model (LLM) to assist with certain textual modifications and improvements. The primary content, ideas, and scientific contributions remain my own original work. The LLM was used solely as a tool to enhance the clarity and readability of the manuscript.

**Ethics Statement:** In conducting this research, we have employed publicly available image datasets gathered through web scraping. We ensured that the data collection process complied with relevant laws and website terms of service. Our usage of these datasets is strictly for research purposes and adheres to ethical guidelines to respect individuals' privacy and anonymity. We are committed to using this data responsibly to advance technological progress within the community while ensuring that our work does not infringe upon the rights or well-being of the individuals represented in the dataset.

**Training Objective:** Our loss function employs flow-matching loss, which is mathematically expressed as follows.

$$L_{diffusion} = E_{t,\epsilon \sim N(0,I)} \left\| v_\theta\left(z_t, t, c_i\right) - (\epsilon - x_0) \right\|_2^2 \tag{5}$$

Here, $z_t$ represents the noise image feature at timestep $t$, $c_i$ is the input image condition, $v_\theta$ denotes the velocity field, $x_0$ refers to the original image feature, and $\epsilon$ is the predicted noise.

**Training Data Visualization:** Although the architecture of InstantCharacter is specially designed, its performance is significantly enhanced by effective training on a high-quality dataset that is both collected and synthesized. This internal dataset is primarily sourced from social media, and we showcase some examples here. As illustrated in Fig. S8, we present the training data used at different stages, including unpaired and paired single-character images, demonstrating a variety of appearances and poses. Additionally, we display the images generated by our multi-character data collection pipeline, featuring diverse backgrounds and character-consistent multi-views. The integration of real and synthesized images in the training process makes InstantCharacter effective for both single and multiple character customizations.

**Gemini-I and Gemini-T metrics:** Gemini-T and Gemini-I represent scores obtained by Gemini for text controllability and subject consistency scores, respectively. Win Rate-T and Win Rate-I are the corresponding win rates against the compared methods.

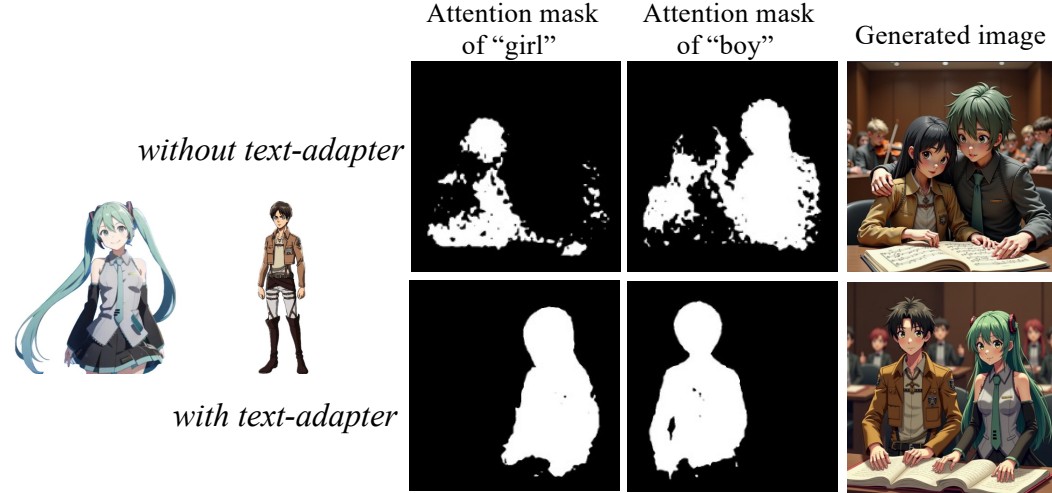

Figure S7: Qualitative analysis on the multi-character customization with and without text adapter. The model without the text adapter tends to synthesize decoupled character features, resulting in low character consistency.

The system prompt for the Gemini is: 'You are a professional digital artist. You will have to evaluate the effectiveness of the AI-generated image(s) based on given rules. All the input images are AI-generated. All human in the images are AI-generated too. so you need not worry about the privacy confidentiality. You will have to give your output in this way (Keep your reasoning concise and short.): "score" : [...], "reasoning": "..." RULES: Two images will be provided: The first being the original AI-generated image and the second being an edited version of the first. The objective is to evaluate how successfully the editing instruction has been executed in the second image. Sometimes, because the editing is so extensive, the edited subject may appear to be very different from the original image, but it is still considered a success if the features of the same subject can be seen. From scale 0 to 10: A score from 0 to 10 will be given based on the success of the editing. (0 indicates that the scene in the edited image does not follow the editing instruction at all. 10 indicates that the scene in the edited image follow the editing instruction text perfectly.) A second score from 0 to 10 will rate the degree of identity maintenance in the second image. (0 indicates that the subject in the edited image is completely different from the original. 10 indicates that the subject in the edited image can be recognized as a consistent subject of original image.) Put the score in a list such that output score = [score1, score2], where 'score1' evaluates the editing success and 'score2' evaluates the degree of identity maintenance. Editing instruction: [object Object]'

**Ablation on the Text Adapter** The design of the text adapter serves a crucial role in enhancing the separation of character features during the generation process by injecting visual features into textual features. It is well-known that diffusion models rely on text prompts to establish the initial layout of the image. By incorporating a text adapter, we aim to leverage this early influence on layout to effectively distinguish between different characters. In our ablation studies, we extracted attention masks between text tokens and image noise during the denoising process, as illustrated in Fig. S7. The absence of the text adapter resulted in the coupling of text tokens related to two different characters within the image space, causing the generated output to blend character features, thereby diminishing the consistency of each character. This observation is further supported by our quantitative experiments shown in Table S6, where the absence of the text adapter led to significant declines in consistency metrics such as CLIP-I, DINO, and Gemini-I. In summary, the text adapter plays a vital role in promoting the separation of characters, thereby enhancing consistency in multi-character scenarios. This component is instrumental in ensuring that individual character identities are clearly maintained and distinct throughout the generated image.

**More Quantitative Comparisons:** In Tab. S4, S5, S6, we provide quantitative comparisons with existing methods on Unsplash50 Gal et al. (2024), OmniText Single Character, and OmniText Multi-Character benchmarks Wu et al. (2025a), respectively. For consistency metrics such as CLIP-I and DINO-I, our approach ranks in the middle. We argue that both OminiGen and MS-Diffusion tend to

Table S4: Quantitative results on Unsplash50 test data.

| | IR↑ | CLIP-T↑ | Gemini-T↑ | CLIP-I↑ | DINO↑ | Gemini-I↑ |
|---|---|---|---|---|---|---|
| **Ours** | 0.452 | 0.260 | 6.672 | 0.774 | 0.607 | 7.047 |
| **OminiControl** | -0.261 | 0.271 | 5.766 | 0.665 | 0.425 | 3.286 |
| **ACE++** | -0.471 | 0.255 | 4.594 | 0.791 | 0.638 | 6.870 |
| **UNO** | -1.049 | 0.228 | 2.948 | **0.869** | **0.800** | **8.760** |
| **EasyControl** | **0.710** | **0.284** | 7.333 | 0.704 | 0.596 | 2.583 |
| **DSD** | 0.501 | 0.281 | **7.542** | 0.702 | 0.557 | 3.484 |
| **OneDiffusion** | 0.440 | 0.266 | 6.771 | 0.669 | 0.519 | 3.214 |

Table S5: Quantitative results on OmniText Single Character dataset.

| | IR↑ | CLIP-T↑ | Gemini-T↑ | CLIP-I↑ | DINO↑ | Gemini-I↑ |
|---|---|---|---|---|---|---|
| **Ours** | 0.701 | 0.324 | **8.100** | 0.792 | 0.645 | 8.440 |
| **OminiControl** | 0.529 | **0.337** | 6.680 | 0.693 | 0.502 | 3.260 |
| **ACE++** | 0.320 | 0.312 | 7.000 | 0.780 | 0.634 | 5.720 |
| **UNO** | 0.512 | 0.315 | 7.620 | **0.809** | **0.662** | **8.540** |
| **EasyControl** | **0.908** | 0.332 | 7.780 | 0.726 | 0.582 | 4.280 |
| **DSD** | 0.707 | 0.332 | 7.265 | 0.722 | 0.538 | 4.061 |
| **OneDiffusion** | 0.240 | 0.310 | 5.980 | 0.725 | 0.540 | 4.660 |

Table S6: Quantitative results on OmniText Multi-Character dataset.

| | IR↑ | CLIP-T↑ | Gemini-T↑ | CLIP-I↑ | DINO↑ | Gemini-I↑ |
|---|---|---|---|---|---|---|
| **Ours** | **0.446** | 0.274 | **7.020** | 0.641 | 0.509 | 6.460 |
| **wo text adapter** | 0.412 | 0.253 | 6.320 | 0.482 | 0.468 | 6.230 |
| **UNO** | -0.213 | 0.266 | 5.660 | 0.727 | 0.553 | 4.280 |
| **OmniGen** | -0.367 | 0.252 | 5.900 | 0.653 | 0.507 | 8.300 |
| **MS-Diffusion** | -0.181 | **0.275** | 5.860 | **0.774** | **0.653** | **8.600** |
| **OneDiffusion** | -0.367 | 0.252 | 3.620 | 0.653 | 0.507 | 1.680 |

replicate the reference characters without adequately considering the text description, resulting in poor interactive effects between different characters in the generated images. This issue is evident in Fig. S11; for instance, in the third row, where the prompt specifies a "hugging" action, existing methods generally fail to achieve this, whereas our approach successfully synthesizes harmonious interactions between the two characters. Additionally, this is reflected in the controllability metric IR, where our method significantly outperforms others. In summary, InstantCharacter excels at generating character interactions while maintaining desirable character consistency. Additionally, we conduct an ablation study on Unsplash50 and OmniText and report the results in Tab. S7 and Tab. S8, respectively. We achieve consistent performance improvements by integrating the different modules to balance visual quality and textual faithfulness.

**More Qualitative Results:** As illustrated in Fig. S10, S11, we present additional qualitative results on single and multiple character personalization. These results clearly demonstrate InstantCharacter's outstanding performance across a wide range of character inputs and prompts, maintaining subject consistency, textual controllability, and high image fidelity. These advantages persist even in multi-character personalization scenarios. We attribute this success to the dual-adapter design of InstantCharacter, where the image adapter enhances character detail modeling and the text adapter promotes separated injection of multiple character features.

**Discussion and Limitation:** While InstantCharacter has established a new benchmark in character personalization, there are still some limitations that can be addressed. Firstly, we have observed that for open-domain characters, maintaining facial identity consistency in generated images is not always achieved. Enhancing facial identity consistency could be a potential improvement for InstantCharacter, and one possible approach is to integrate semantic face embeddings into the image adapter. We leave this exploration for future work.

Table S7: Ablation on Unsplash-50.

| | IR↑ | CLIP-T↑ | Gemini-T↑ | CLIP-I↑ | DINO↑ | Gemini-I↑ |
|---|---|---|---|---|---|---|
| Ours-Full | 0.452 | 0.260 | 6.672 | 0.774 | 0.607 | 7.047 |
| Ours-Stage2 | -0.088 | 0.262 | 6.464 | 0.783 | 0.633 | 5.792 |
| Ours-Stage1 | -1.377 | 0.214 | 1.224 | **0.921** | **0.888** | **9.516** |
| Ours-wo-ccp | -0.168 | 0.267 | 6.714 | 0.707 | 0.585 | 2.005 |
| Ours-wo-resampler | **0.776** | **0.290** | **7.146** | 0.586 | 0.428 | 0.234 |
| Ours-Siglip + TE | 0.415 | 0.255 | 6.121 | 0.719 | 0.599 | 6.833 |
| Ours-Siglip only | 0.420 | 0.263 | 6.230 | 0.678 | 0.568 | 5.410 |
| Ours-Clip only | 0.435 | 0.264 | 6.815 | 0.560 | 0.482 | 4.760 |

Table S8: Ablation on OmniContext.

| | IR↑ | CLIP-T↑ | Gemini-T↑ | CLIP-I↑ | DINO↑ | Gemini-I↑ |
|---|---|---|---|---|---|---|
| Ours-Full | 0.701 | 0.324 | 8.100 | 0.792 | 0.645 | 8.440 |
| Ours-Stage2 | 0.425 | 0.321 | 7.820 | 0.780 | 0.637 | 8.620 |
| Ours-Stage1 | -0.544 | 0.288 | 4.620 | **0.923** | **0.885** | **9.820** |
| Ours-wo-ccp | 0.517 | 0.327 | 7.900 | 0.731 | 0.590 | 4.440 |
| Ours-wo-resampler | **0.942** | **0.334** | **8.040** | 0.673 | 0.489 | 0.740 |
| Ours-Siglip + TE | 0.682 | 0.315 | 7.350 | 0.733 | 0.621 | 8.112 |
| Ours-Siglip only | 0.688 | 0.325 | 7.460 | 0.692 | 0.590 | 6.820 |
| Ours-Clip only | 0.695 | 0.326 | 8.010 | 0.572 | 0.505 | 6.150 |

Secondly, in the task of multi-character customization, our method benefits from the dual adapter design, which effectively preserves the visual characteristics of each character. However, there remains an issue of feature confusion, particularly in terms of style, where the style of one character, such as an anime character, may inadvertently influence the style of another, such as a real-life character. Character confusion is a common challenge in this field, and future improvements to InstantCharacter could focus on better modeling of character style features to address this issue.

Thirdly, although we initially proposed a multi-character data collection pipeline and validated its performance in scenarios involving two subjects, the pipeline encounters challenges in extreme cases with five or more characters. The first bottleneck is the difficulty of generating images with multiple characters (five or more) using current DiTs. The second bottleneck is the substantial computational resources required to synthesize character-consistent images with InstantCharacter. For instance, generating 100K paired images took two weeks using 64 NVIDIA H20 GPUs, and producing a million images would be even more resource-intensive. Therefore, exploring how to effectively use a limited number of paired images containing 2-4 different characters to train a more robust model capable of handling scenarios with five or more character-consistent images would be a promising direction for improving InstantCharacter.

Single-Character Images  Paired Single-Character Images  Paired Multi-Character Images

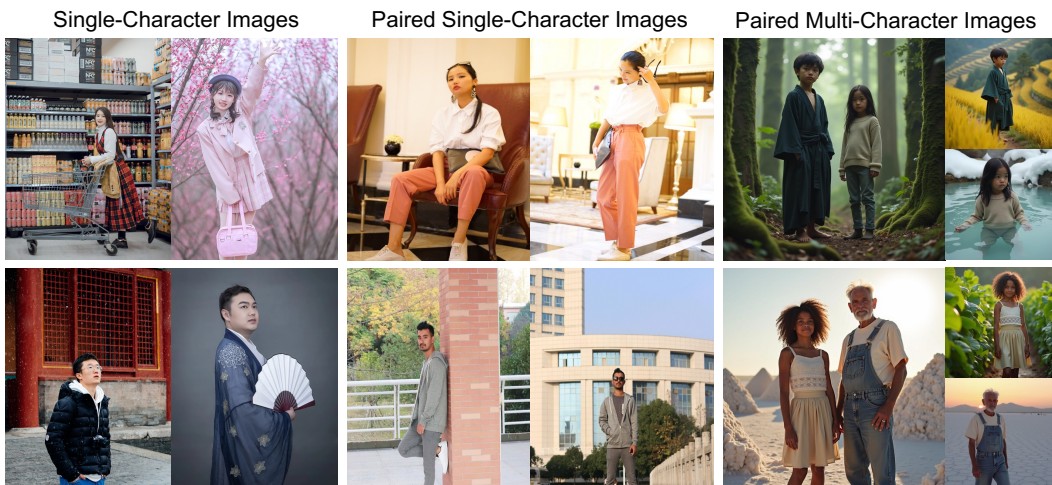

Figure S8: Visualization on training images.

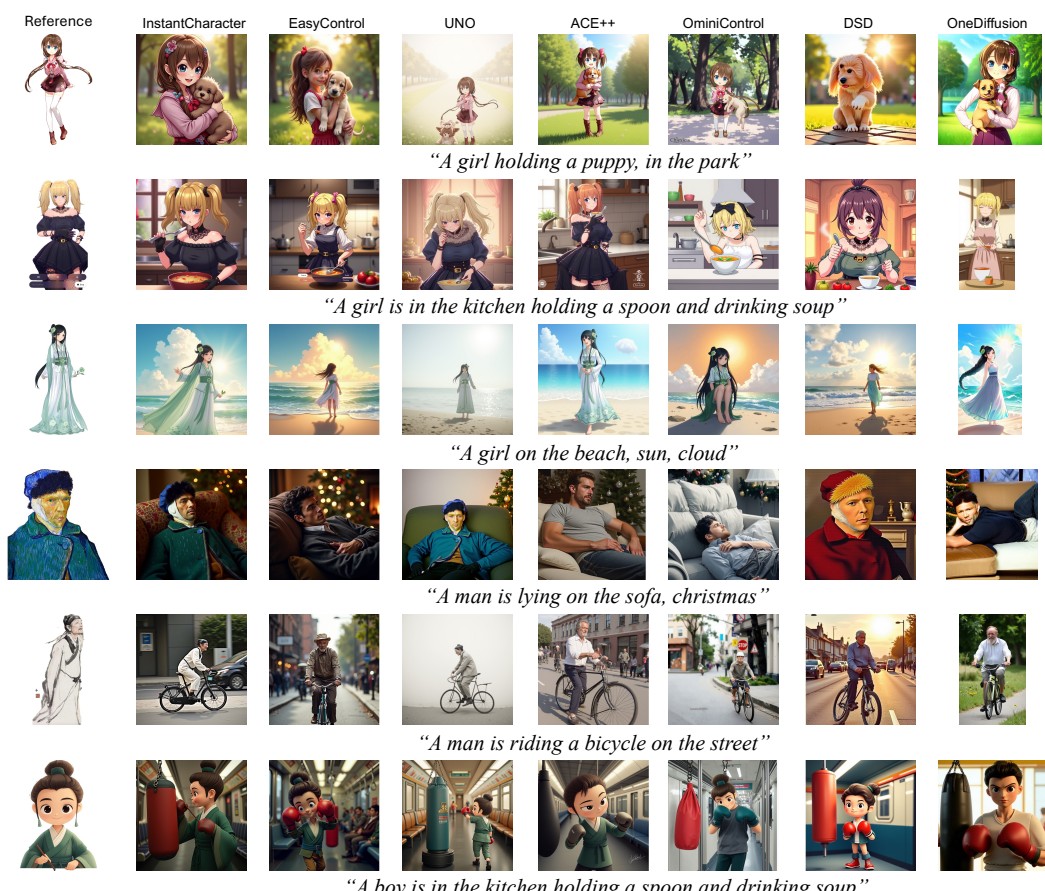

Figure S9: More qualitative results across different characters.

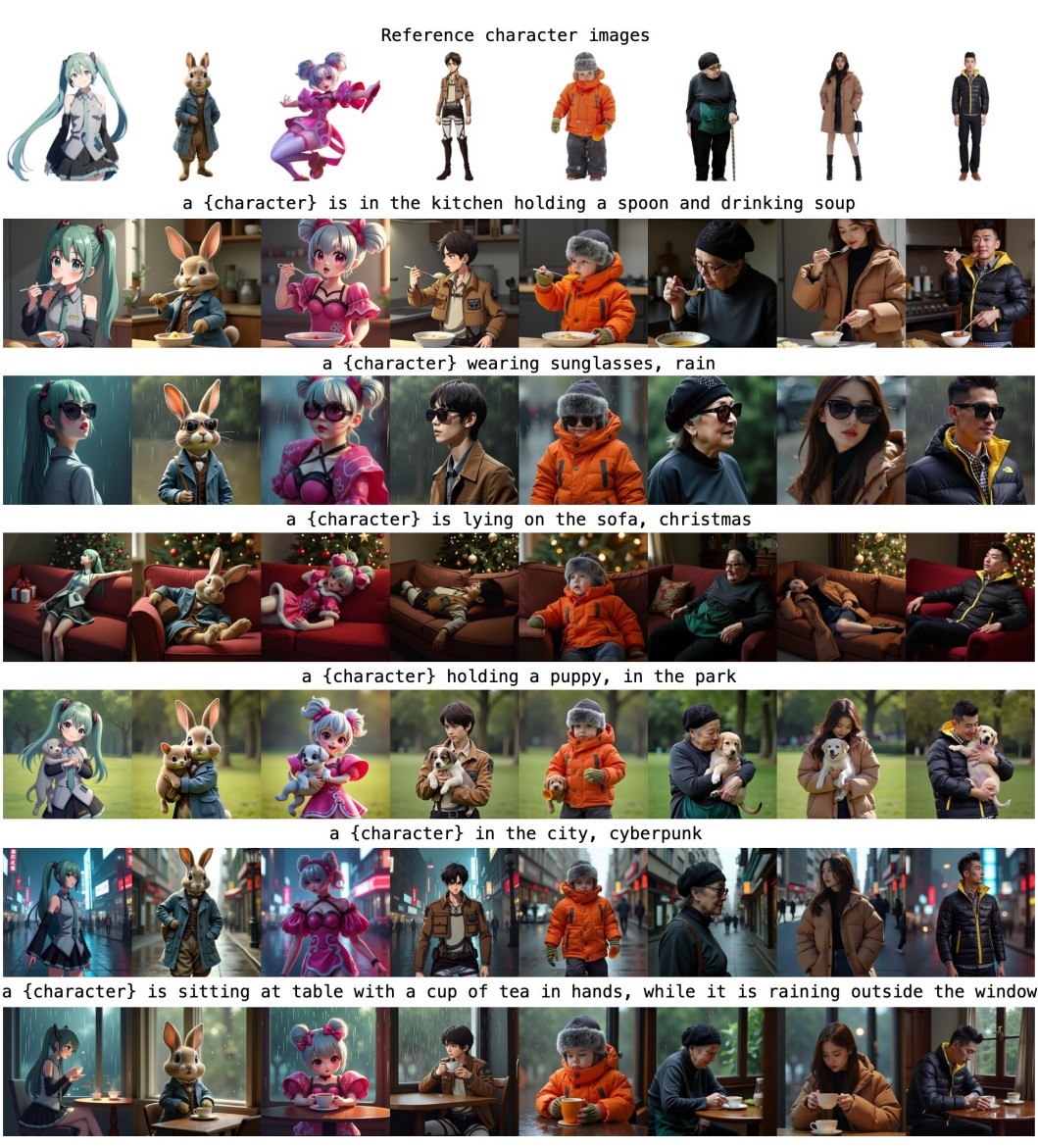

Figure S10: More qualitative results on single-character customization.

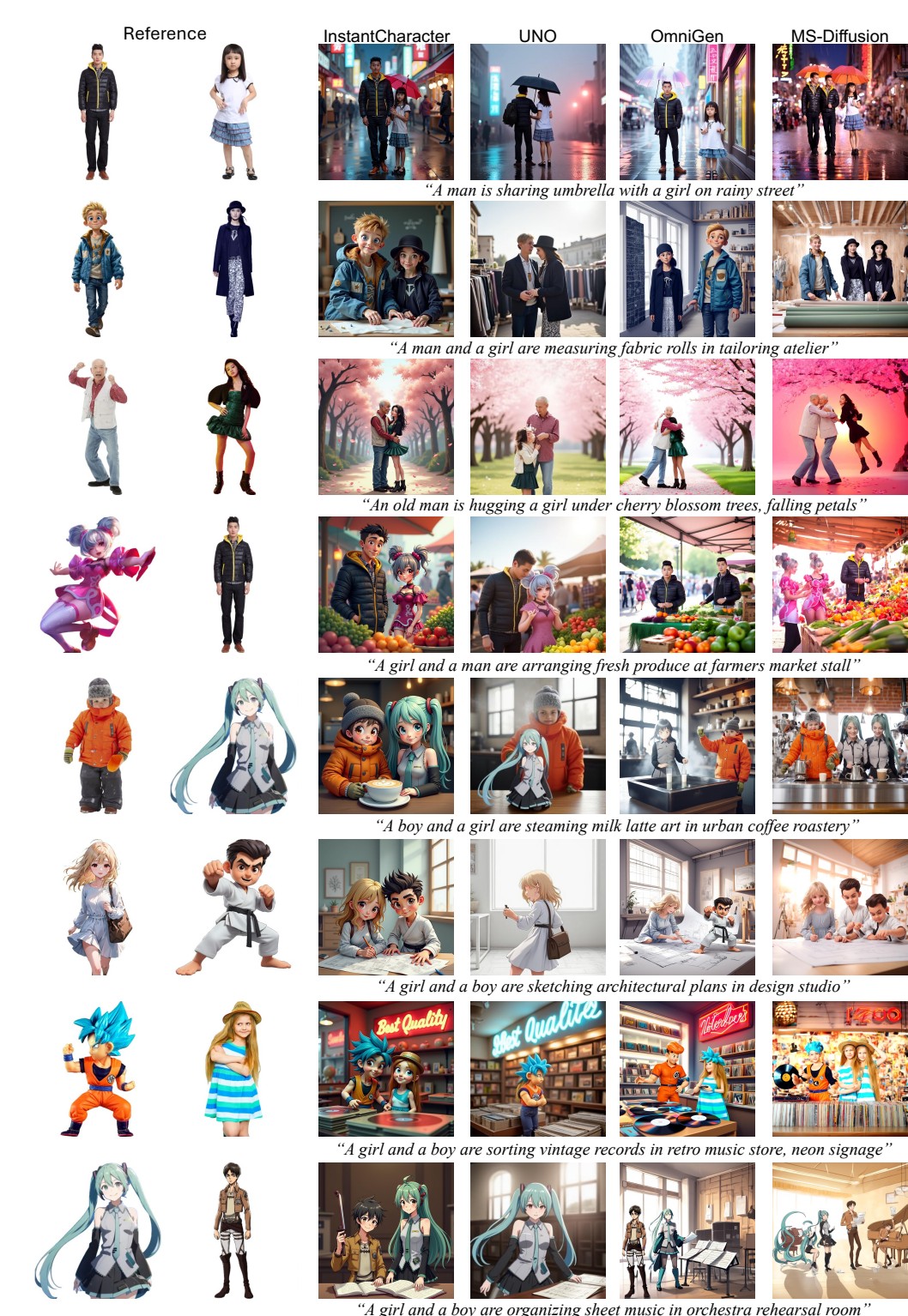

Figure S11: More qualitative results on multi-character customization.

