# OpenReview forum: "InstantCharacter: Personalize Any Characters with a Scalable Diffusion Transformer Framework"
_ICLR.cc/2026/Conference — Submitted to ICLR 2026_

### Official Review · Reviewer_8u1R · 2025-10-18

**Soundness:** 2
**Presentation:** 2
**Contribution:** 2
**Rating:** 4
**Confidence:** 4

**Summary:**

The authors of the paper propose a solution for customized text-to-image generation.
They leverage adapter training on custom data and leave the diffusion network untouched.
There are multiple adapters for reference image and text processing, which come from preselected image and text feature extraction models. Only adapters get trained, and everything else remains untouched. They propose 3 stages of the training process, which have different purposes. The authors claim that their method can generate multiple characters as well.

**Strengths:**

1. Constructing a custom dataset with 10 million examples for character customization
2. Adapter-based text-to-image customization approach, which is flexible and avoids touching the diffusion model or causing knowledge loss.

**Weaknesses:**

1. Poor results on almost all quantitative comparisons. The superiority of the proposed model is not justified with quantitative comparisons. The results are mostly worse than other competitive solutions. This also questions the fairness of the example selection of qualitative comparisons.

   Specifically, in the "Quantitative Results" section, the claim about UNO performance is noticeably poor. To justify UNO's better performance in Tab 1 and Tab 2, the authors refer to a qualitative comparison. For a fair comparison, the claim and the justification should be in the same field: either both should be compared in qualitative results or in quantitative results.

2. Limited scientific novelty. Even though the authors have done an extensive job, there is nothing unique or new.

3. The paper is poorly written.

    a) In expression 1, there is F and F^{Q}, but in the description, F is explained as a concatenation result of F^{siglip}, F^{dino}; meanwhile, F is the output of the attention.

    b) (small note) In expressions 2 and 3, H is noted as the hidden features of DiT. However, in expression 4, H is for text embeddings.

    c) In section 3.1.1 (208-211), the authors mention that the output features of the image encoder, F^{siglip}_{l} and F^{Dino}_{r}, go through separate encoders (multiple encoders) for further processing. What encoders are they? It can't be the image adapter, cause the authors mention separate and multiple encoders. There are no details about them.

**Questions:**

1. Would you give more details about "learnable queries" you mentioned in multiple places (e.g., 236)?
2. In the ablation study, how are reference image features used when the Transformer Encoder is removed?
3. In Figure 2, how are the hidden features of DiT described as image and textual features, in 2 separate groups? Aren't the Text Adapter outputs and textual embeddings supposed to have an attention before being injected into DiT?

---

> ### Author Response · Authors · 2025-11-21
> **Part 1**
>
> - **Poor results on almost all quantitative comparisons. This also questions the fairness of the example selection of qualitative comparisons. Specifically, in the "Quantitative Results" section, the claim about UNO performance is noticeably poor. To justify UNO's better performance in Tab 1 and Tab 2, the authors refer to a qualitative comparison. For a fair comparison, the claim and the justification should be in the same field: either both should be compared in qualitative results or in quantitative results.**
>
> We appreciate this insightful comment. Regarding the quantitative comparisons, we would like to clarify the inherent trade-off in current metrics and justify our superiority through Human Evaluation (User Study).
>
> Most identity preservation metrics (e.g., CLIP-I, DINO) calculate the cosine similarity between embeddings. As a result, methods like UNO can achieve artificially high scores by exhibiting a "copy-paste" effect—essentially reconstructing the reference image rather than following the editing instructions. However, this comes at a significant cost: as shown in the tables, their text controllability metrics drop strictly.
>
> This is why we incorporate qualitative results—not to replace quantitative data, but to interpret it. Visualizations reveal that while competitors score high on ID, they fail to generate the target content defined by the prompt.
>
> To address the concern of fairness and provide an objective quantitative standard, we included a User Study in Table 1. In this human-perceptual metric—which is often considered the gold standard for generative tasks—our method consistently outperforms competitors. This aligns with our qualitative results and confirms that our method achieves the best balance between identity preservation and text-driven editability.
>
> Thanks for pointing out the question again, and please also refer to the similar question in the Part 1 response of Reviewer UcZj.
>
>
> - **Limited scientific novelty. Even though the authors have done an extensive job, there is nothing unique or new.**
>
> We respectfully clarify that our approach offers significant scientific novelty in the domain of character customization, addressing several critical challenges with targeted solutions. Our method is not merely an extensive application of existing techniques; instead, it innovatively tackles the issues of character consistency, text controllability, and enhanced visual quality through uniquely designed components.
>
> The multi-granularity feature extraction (region level and low level) is tailored to capture intricate character features at various levels, ensuring fine-grained consistency that traditional methods struggle to achieve.
>
> Furthermore, the scalable image adapter architecture ensured that the fine-grained features could be effectively accommodated in the latent space of the DiT model. Our framework extends beyond single-character scenarios with the introduction of a multi-character data construction pipeline and a Dual-Adapter Architecture, facilitating effective multi-character personalized generation.
>
> Additionally, the three-stage training strategy, integrating unpaired, paired, and high-resolution data, strategically balances text controllability with visual quality, refining the model at each stage.
>
> These innovations collectively form a robust and scalable solution for single/multi-character customization, clearly distinguishing our work as a novel and meaningful contribution to the research community.
>
> - **The paper is poorly written.**
>
> **a) In expression 1, there is F and F^{Q}, but in the description, F is explained as a concatenation result of F^{siglip}, F^{dino}; meanwhile, F is the output of the attention.**
>
> **b) (small note) In expressions 2 and 3, H is noted as the hidden features of DiT. However, in expression 4, H is for text embeddings.**
>
> **c) In section 3.1.1 (208-211),  F^{siglip}{l} and F^{Dino}{r}, go through separate encoders (multiple encoders) for further processing. What encoders are they?.**
>
> We sincerely apologize for the confusion caused by the notation inconsistencies and unclear descriptions.  We will revise the manuscript as follows:
>
> 1) We will correct the notation to distinguish the features, where we denote the concatenated features (SigLIP and DINO) as F, and the output of the attention module as $A$ to avoid symbol overloading.
>
> 2) $H$ remains the hidden features of DiT, and we will rename the text embeddings in Eq. 4 to $C$  (Context) to ensure consistency.
>
> 3) We clarify that the "separate encoders" mentioned in the text actually constitute the internal structure of our Image Adapter, not external modules. As illustrated in the Figure 2, the Image Adapter consists of two parallel Transformer blocks. These parallel branches are explicitly designed to capture specific feature granularities: one specializes in low-level features while the other focuses on region-level features.

---

> ### Author Response · Authors · 2025-11-21
> **Part 2**
>
> - **Would you give more details about "learnable queries" you mentioned in multiple places (e.g., 236)?**
>
> Specifically, we employ a set of 1,024 randomly initialized learnable queries. These queries interact with the visual features extracted from the Image Adapter via a cross-attention mechanism. Importantly, this fusion is timestep-aware, allowing the model to dynamically adjust the visual guidance at different denoising stages, thereby significantly enhancing character consistency.
>
> We also verified this design choice through ablation studies on the number of queries (comparing 64, 512, and 1024). Our empirical results demonstrated that the setting of 1,024 yields superior performance in balancing identity preservation and generation quality.
>
> - **In the ablation study, how are reference image features used when the Transformer Encoder is removed?**
>
> In the ablation setting where the Transformer Encoder (TE) is removed, we directly feed the concatenated features extracted from SigLIP and DINO into the Time Resampler. We appreciate the reviewer for pointing this out, and we will explicitly clarify this configuration in the revised manuscript.
>
> - **In Figure 2, how are the hidden features of DiT described as image and textual features, in 2 separate groups? Aren't the Text Adapter outputs and textual embeddings supposed to have an attention before being injected into DiT?**
>
> We appreciate your attention to these architectural details. In the Flux.1 framework, image and text tokens are concatenated, allowing us to treat them as separate groups within the sequence. Furthermore, as you correctly noted, we apply cross-attention between the Text Adapter outputs and the textual embeddings, using the result to replace the original embeddings. We apologize if Figure 2 was ambiguous; we will refine the figure in the revision to explicitly show the cross-attention block and the token concatenation structure.

---

### Official Review · Reviewer_UcZj · 2025-10-27

**Soundness:** 3
**Presentation:** 3
**Contribution:** 2
**Rating:** 6
**Confidence:** 4

**Summary:**

This paper proposes a scalable framework, called InstantCharacter, for character customization built upon a DiT-based diffusion model: Flux. InstantCharacter consists of three key components: a scalable dual-adapter architecture that parses character features and interacts with DiTs latent space, a progressive three-stage training strategy that separates training for character consistency, text editability, and visual fidelity, and a new pipeline for constructing training data pairs for multi-character customization.

**Strengths:**

1. The images generated by the proposed method for character customization are plausible and impressive.
2. This paper is well-written and well-organized.
3. This paper provides a versatile 10-million-level character dataset, which contains paired (multi-view character) and unpaired (text-image combinations) subsets.
4. Extensive experiments are conducted to evaluate the performance of the proposed method.

**Weaknesses:**

1. What are the objective loss functions used in the three training stages of this paper? The first stage involves the reconstruction of the input image, which is presumably achieved using standard diffusion loss. However, both the second and third stages involve transformations of the original image; what loss functions are used in these stages?

2. As shown in Tables 1 and 2, the quantitative results of the proposed method do not seem very satisfactory, as it fails to demonstrate a clear advantage over existing baseline methods.

3. The ablation studies conducted in this paper are not sufficient, and the effects of many important components have not been evaluated. For example, what would be the difference in performance with and without the text adapter? What would be the difference in performance with and without the dual-stream feature fusion strategy (Section 3.1.1)?

4. What is the time efficiency of different methods? Some evaluations regarding this should be conducted.

**Questions:**

Please see **Weaknesses**.

Others:

Does the proposed method in this paper support 3-character(or more) personalization? Currently, many existing methods do not limit the number of concepts when performing multi-concept personalization.

---

> ### Author Response · Authors · 2025-11-21
> **Weakness 1-2**
>
> - **What are the objective loss functions used in the three training stages of this paper?**
>
> Sorry for the confusion. As introduced in line 631 of the appendix, all three stages use the flow matching Loss. While the input conditions change on the second and third stages, the training objective remains supervised by the target ground-truth images using the standard flow matching objective, ensuring consistent optimization behavior.
>
> - **The quantitative results of the proposed method do not seem very satisfactory**
>
> We understand the reviewer's concern regarding the quantitative landscape. To better interpret these results, it is crucial to first clarify the nature of the six independent metrics used in our evaluation.
>
> Textual controllability: IR (using the ImageReward model) and CLIP-T (similarity computed by text and image CLIP embeddings) measure how well the image follows the prompt.
>
> Subject consistency: CLIP-I (CLIP image embedding similarity) measures fidelity to the reference subject.
> VLM-based evaluation: Gemini-T and Gemini-I utilize the Gemini VLM to assess instruction following and identity preservation, respectively.
>
> 1. Metric Independence & The "Copy-Paste" Trap:
> As commonly observed in recent literature [1, 2], these automated metrics are completely decoupled. A high score in one dimension does not imply overall quality. In fact, they often negatively correlate. For instance, baselines like UNO achieve high CLIP-I/Gemini-I scores by overfitting to the reference image (a "copy-paste" effect). While this boosts ID metrics, it inevitably causes a collapse in IR/CLIP-T scores because the model refuses to execute the text edits.
>
> 2. Pushing the Pareto Frontier:
>
> Consequently, the perceived lack of a dominant advantage in individual columns stems from the polarized performance of baselines—they tend to occupy extreme ends of the spectrum (high ID/low Edit or high Edit/low ID).
> In contrast, our method strikes a superior balance. While we may not surpass the specialized baselines in every single metric, we effectively push the Pareto frontier of these conflicting objectives. Our method achieves a comprehensive improvement that single-dimensional metrics fail to capture fully.
>
> 3. Qualitative & Human Evaluation:
> We emphasize that our qualitative results (shown in Figure 4) are not cherry-picked but represent the general capability of our model to succeed in complex scenarios where baselines fail. Most importantly, the User Study—which evaluates the joint quality of identity and editability—confirms our superiority, validating that our "balanced" approach aligns best with human preference.
>
> [1] An Image is Worth One Word: Personalizing Text-to-Image Generation using Textual Inversion
>
> [2] Diffusion Self-Distillation for Zero-Shot Customized Image Generation

---

> ### Author Response · Authors · 2025-11-21
> **Weakness 3-5**
>
> - **The ablation studies on the text adapter and dual-stream feature fusion strategy (Section 3.1.1)**
>
> Thank you for pointing out the need to further demonstrate the role of the Text Adapter and fusion strategy.
>
> The design of the text adapter serves a crucial role in enhancing
> the separation of character features during the generation process by injecting visual features into
> textual features. It is well-known that diffusion models rely on text prompts to establish the initial
> layout of the image. By incorporating a text adapter, we aim to leverage this early influence on
> layout to effectively distinguish between different characters. In our ablation studies, we extracted
> attention masks between text tokens and image noise during the denoising process, as illustrated
> in Fig. S7. The absence of the text adapter resulted in the coupling of text tokens related to two
> different characters within the image space, causing the generated output to blend character features,
> thereby diminishing the consistency of each character. This observation is further supported by our
> quantitative experiments shown in Table S6, where the absence of the text adapter led to significant
> declines in consistency metrics such as CLIP-I, DINO, and Gemini-I.
>
> We have updated the analysis, table, and figure content in the appendix, in lines 665, 686, and 727.
>
> |                      | **IR↑** | **CLIP-T↑** | **Gemini-T↑** | **CLIP-I↑** | **DINO↑** | **Gemini-I↑** |
> |----------------------|---------|-------------|---------------|-------------|-----------|---------------|
> | **Ours**             | 0.446   | 0.274       | 7.020         | 0.641       | 0.509     | 6.460         |
> | **wo text adapter**  | 0.412   | 0.253       | 6.320         | 0.482       | 0.468     | 6.230         |
>
> As for the fusion strategy, we have conducted ablation results in Tables S7 and S8 in the appendix. As illustrated in lines 729-730 and lines 740-741, the variant "Ours-Siglip only" represents the model without the fusion strategy, the variant "Ours-Siglip + TE"
> represents the model with the dual feature fusion strategy (Siglip and DINO feature). It can be clearly observed that the consistency metrics like CLIP-I, DINO, and Gemini-I are considerably improved by the fusion strategy, which validates our design motivation.
>
> Sorry for the confusion on the ablation variant names. We will clarify this in the later version.
>
> - **time efficiency of different methods**
>
> We appreciate the reviewer's suggestion regarding efficiency evaluation.
>
> Since our method is built upon Flux.1-dev, the inference latency is primarily determined by the backbone model itself (which takes approx. 14s for a standard generation process on an NVIDIA H20 GPU). We have conducted a comparative analysis of inference times across different methods under the same environment:
>
>  | Method | Backbone | Inference Time (s) |
>  | :--- | :--- | :---: |
>  | OminiControl | Flux.1-schnell | 8s |
>  | ACE++ | Flux.1-dev | 15s |
>  | EasyControl | Flux.1-dev | 16s |
>  | **Ours** | Flux.1-dev | 18s |
>  | DSD | Flux.1-dev | 20s |
>  | OneDiffusion | Next-DiT | 22s |
>  | UNO | Flux.1-dev | 35s |
>
> As shown, our method adds only a marginal overhead (~4s) to the base model for feature injection. We are significantly faster than complex pipelines like UNO (35s) and comparable to other state-of-the-art DiT-based approaches (e.g., ACE++, EasyControl). We will include this comparison in the final revision.
>
> - **3-character(or more) personalization? Currently, many existing methods do not limit the number of concepts when performing multi-concept personalization.**
>
> Thank you for the valuable question.Firstly our framework inherently supports personalization for 3 or more characters. As detailed in the paper, our data pipeline and method implementation are designed to be generic. Architecturally, extending the model to handle more subjects is straightforward, provided that corresponding multi-subject training data can be synthesized or collected.
>
> Secondly, while the technical implementation of supporting multi-concept personalization is feasible in some existing methods, achieving robust generalization for complex interactions among multiple characters (e.g., 3+) remains a broader challenge in the field. Our framework is designed with this scaling potential in mind and can be adapted to such scenarios given sufficient data. We leave this to our future work.

---

### Official Review · Reviewer_yUPG · 2025-10-31

**Soundness:** 2
**Presentation:** 2
**Contribution:** 2
**Rating:** 4
**Confidence:** 3

**Summary:**

This paper presents InstantCharacter, a novel and scalable framework for character customization built upon a foundation Diffusion Transformer (DiT). This work effectively addresses the critical gap left by previous U-Net and optimization-based methods, which suffered from limited generalization and compromised textual controllability. The core technical contributions include a scalable dual-adapter architecture for injecting character-specific features and enhancing multi-subject layout control , complemented by an effective three-stage progressive training strategy. Furthermore, the authors construct a large-scale (10-million-level) character dataset for training the framework. Experimental results demonstrate superior performance in generating high-fidelity, text-controllable, and character-consistent images across diverse appearances and styles.

**Strengths:**

* The work is the first to develop a DiT-based framework specifically optimized for character customization, introducing a novel dual-adapter design (Image Adapter and Text Adapter) that seamlessly interacts with the DiT's latent space to maintain high-fidelity results.
* The proposed three-stage progressive training strategy is highly effective in accommodating the heterogeneous 10M dataset, successfully decoupling the training for character consistency, textual controllability, and image fidelity.
* The comparative experiments demonstrate the method's superior capabilities in consistently preserving character identity and high facial fidelity while maintaining precise text controllability, showing excellent potential for real-world applications.
* Introducing the new Character350 evaluation benchmark.

**Weaknesses:**

The paper repeatedly emphasizes the “scalability” of its framework, yet only briefly mentions its Transformer-based adapter design.
However, the paper lacks a rigorous technical argument or experimental evidence to convincingly justify why this DiT-based dual-adapter approach holds a tangible advantage over U-Net-based adapters or other micro-tuning techniques specifically when scaling to significantly larger DiT models. This central claim requires more thorough substantiation

**Questions:**

*  The 10-million-level dataset is a fundamental component of the model's success. It is highly recommended that the authors provide a more detailed and comprehensive explanation in the main text or supplementary material regarding the dataset's construction, cleaning, and filtering standards, as this information is crucial for reproducibility and understanding the model's performance.
* Please unify the formatting of the first column across all tables in the paper (e.g., consistently bold or consistently non-bold) for improved visual consistency.

---

> ### Author Response · Authors · 2025-11-21
>
> - **The paper repeatedly emphasizes the “scalability” of its framework, yet only briefly mentions its Transformer-based adapter design. However, the paper lacks a rigorous technical argument or experimental evidence to convincingly justify why this DiT-based dual-adapter approach holds a tangible advantage over U-Net-based adapters or other micro-tuning techniques specifically when scaling to significantly larger DiT models. This central claim requires more thorough substantiation**
>
> We thank the reviewer for this critical question. Our claim regarding "scalability" is substantiated by our injection mechanism and the extensible nature of our adapter architecture.
>
> 1) Unlike complex hypernetworks or intrusive modifications, our adapter is designed with high adaptability. Technically, the adapter's output is directly added to the features of the DiT backbone immediately after the original attention calculation. This "additive" strategy ensures seamless integration into the backbone’s residual stream without disrupting its internal feature topology, making it inherently scalable to DiT models of any size.
>
> 2) Furthermore, our Transformer-based design allows for arbitrary scaling. As the parameter size of the DiT backbone increases, we can easily scale up the adapter (e.g., by stacking more layers) to match the required representational capacity, ensuring sufficient guidance for larger models.
>
> 3) To empirically validate this scalability, we conducted an ablation study by increasing the adapter's capacity from 1 to 4 layers on the current dataset. The results demonstrate a clear scaling schema: increasing the adapter's depth significantly boosts visual fidelity metrics (CLIP-I, DINO, Gemini-I) without sacrificing much text controllability. For instance, Gemini-I improves from 7.920 (1 Layer) to 8.300 (4 Layers). This confirms that our architecture effectively leverages increased parameters to capture finer details, justifying the "scalability" of our DiT-based dual-adapter design.
>
> | Transformer Layers | IR↑ | CLIP-T↑ | Gemini-T↑ | CLIP-I↑ | DINO↑ | Gemini-I↑ |
> | :--- | :---: | :---: | :---: | :---: | :---: | :---: |
> | 1 Layer | **1.025** | **0.314** | **9.245** | 0.765 | 0.572 | 7.920 |
> | 2 Layers | 1.006 | 0.305 | 9.137 | 0.786 | 0.593 | 8.165 |
> | 4 Layers (Full)| 0.994 | 0.308 | 9.182 | **0.795** | **0.604** | **8.300** |
>
> - **The 10-million-level dataset is a fundamental component of the model's success. It is highly recommended that the authors provide a more detailed and comprehensive explanation in the main text or supplementary material regarding the dataset's construction, cleaning, and filtering standards, as this information is crucial for reproducibility and understanding the model's performance.**
>
> Our dataset collection process involves minimal cleaning and filtering strategies. It simply use full-body photos of real or anime characters without requiring extensive preprocessing, aside from face detection. For training single-character InstantCharacter models, we apply a straightforward filtering strategy to exclude images containing multiple faces, which significantly expands our dataset.
>
> - **Please unify the formatting of the first column across all tables in the paper (e.g., consistently bold or consistently non-bold) for improved visual consistency.**
>
> Thank you for the suggestion, we will correct them in the refined version.

---

### Official Review · Reviewer_3n5x · 2025-11-01

**Soundness:** 2
**Presentation:** 2
**Contribution:** 2
**Rating:** 4
**Confidence:** 4

**Summary:**

This paper proposes "InstantCharacter," a framework for character customization built on the FLUX.1 diffusion transformer (DiT) backbone. It introduces a dual-adapter architecture to address the limitations of prior U-Net and tuning-based methods. An Image Adapter injects multi-level character features (from SigLIP and DINOv2) into the DiT's image tokens to ensure character consistency. Concurrently, a Text Adapter injects character features into the text tokens, which is claimed to improve layout control, especially for multi-character scenarios. The method is trained on a massive 10-million sample dataset using a progressive three-stage strategy to balance consistency, controllability, and image fidelity.

**Strengths:**

1. Significant Engineering Effort: The authors demonstrate a substantial engineering effort, including the curation of a massive 10M-sample dataset, the implementation of a complex multi-stage/multi-resolution training pipeline, and a novel synthetic data-generation loop for multi-character images.

1. Thoughtful Architecture for DiT: The design of the Image Adapter, which uses stacked transformers to process multi-level features (low-level, region-level, and semantic) from multiple encoders (SigLIP + DINOv2), is a thoughtful approach to capturing a robust character representation suitable for the DiT.

1. Multi-Character Handling: The explicit inclusion of a Text Adapter to manage multi-character generation is a valuable design choice. This design choice directly addresses a common failure point in personalization models.

**Weaknesses:**

1. Outdated Premise and Inaccurate Framing: The paper's primary motivation, posing itself as a superior alternative to U-Net-based adapters, is largely outdated. The SOTA research frontier has decisively shifted to DiT-based methods for some time. This inaccurate framing extends to its claims of being the "first DiT-based framework", which is factually contradicted by the paper's own citations and comparisons to other concurrent DiT-based methods .

2. Incremental Contribution: Viewed in the correct context (as one of many DiT-adapter methods), the methodological novelty is limited. The dual-adapter approach (Image + Text adapters) is a logical, but not highly innovative, recombination of existing concepts (e.g., IP-Adapter's cross-attention injection, PhotoMaker's fused embeddings) applied to a DiT.

3. Critically Incomplete Baseline Comparisons: The comparisons do not reflect the true SOTA and are missing the actual competitors.

- It omits the dominant U-Net/SDXL-based SOTA methods that set the community benchmark, namely InstantID and PhotoMaker. A SOTA claim is impossible without comparing to them.

- More importantly, it fails to compare against or even acknowledge other advanced DiT-native personalization methods (e.g., FLUX-Kontext), which represent the true state-of-the-art for this backbone. This makes the paper's performance unevaluated against its true peers.

4. Incomplete and Ambiguous Ablation Study: The paper's core contribution is its "dual-adapter" architecture, but the ablation study fails to scientifically validate this specific design choice.

- The Text Adapter's contribution is unproven. The paper claims the Text Adapter is crucial for multi-character layout and separation . To prove this, an ablation study w/o Text Adapter should have been run.

5. Unsatisfactory Qualitative Results & Model Bias: Despite claims of high fidelity, the qualitative results in the appendix (Figure S9) reveal significant failures in ID consistency. The model shows a strong stylistic bias. For example, when given 2D cartoon characters (e.g., columns 1 and 3) and the prompt "a {character} wearing sunglasses, rain", the generated outputs are rendered as 3D realistic characters, and the clothing is noticeably changed. This demonstrates a failure to preserve the core style and details of the reference character, undermining the paper's central claims.

6. Dependence on Proprietary Data: The method's performance is inextricably linked to a massive, 10-million-sample proprietary dataset. This makes the results non-reproducible and makes it impossible to disentangle the contribution of the architecture from the contribution of the data.

**Questions:**

Please refer to Weakness section. I will consider raise my score if all my concerns are well addressed.

---

> ### Author Response · Authors · 2025-11-21
> **Weakness 1-2**
>
> - **Outdated Premise and Inaccurate Framing: The paper's primary motivation, posing itself as a superior alternative to U-Net-based adapters, is largely outdated. The SOTA research frontier has decisively shifted to DiT-based methods for some time. This inaccurate framing extends to its claims of being the "first DiT-based framework", which is factually contradicted by the paper's own citations and comparisons to other concurrent DiT-based methods.**
>
> Thank you for the comments, and we will address your concerns as follows:
>
> Acknowledgment of concurrent work: We acknowledge that several contemporary studies are exploring DiT-based methods, and infact most of them are what we compared in the main paper,  including the works like DSD[1], OneDiffusion[2], OminiControl[3], ACE++[4], UNO[6], and EasyControl[5]. We have discussed and compared extensively in our paper, as introduced in line 329 in the main paper. As noted in our qualitative and quantitative experiments, our framework exhibits exceptional performance compared to other DiT-based methods. We have provided comprehensive results and analyses that highlight the strengths and advantages of our approach within the DiT paradigm, further supporting our contribution as a noteworthy advance.
>
> Positioning of our work: While we recognize that the research frontier has indeed shifted towards DiT-based models, our paper aims to contribute significantly to this area by providing a novel framework that enhances the capabilities of DiT-based methods. We shoule claim that our primary motivation was not merely to offer an alternative to U-Net-based adapters but rather to push the boundaries within DiT-based methodologies.
>
> [1] Diffusion Self-Distillation for Zero-Shot Customized Image Generation, CVPR2025
>
> [2] One Diffusion to Generate Them All, CVPR2025
>
> [3] OminiControl: Minimal and Universal Control for Diffusion Transformer, ICCV2025
>
> [4] ACE++: Instruction-Based Image Creation and Editing via Context-Aware Content Filling, ICCV2025 WorkShop
>
> [5] EasyControl: Adding Efficient and Flexible Control for Diffusion Transformer, ICCV2025
>
> [6] Less-to-More Generalization: Unlocking More Controllability by In-Context Generation, ICCV2025
>
>
> - **Incremental Contribution: Viewed in the correct context (as one of many DiT-adapter methods), the methodological novelty is limited. The dual-adapter approach (Image + Text adapters) is a logical, but not highly innovative, recombination of existing concepts (e.g., IP-Adapter's cross-attention injection, PhotoMaker's fused embeddings) applied to a DiT.**
>
> We would like to address the raised points and clarify specific aspects of our work:
>
> Framework's efficacy and performance: While our method, along with others, falls within the DiT-adapter framework, it is specifically tailored for character customization scenarios. Our experimental results clearly demonstrate its effectiveness and distinctiveness in achieving superior performance compared to other contemporary DiT-based methods. This positions our framework as a significant advancement within the character customization context.
>
> Clarifications on methodological novelty: It is important to delineate that our framework's innovation lies not merely in the recombination of existing concepts, but in their strategic integration and enhancement. Specifically, as illustrated in Figure 2, the cross-attention injection follows the Image-Adapter process and isn't part of the Image-Adapter itself. Our Image-Adapter was conceived to address two key motivations:
>
> a) The inadequacies of traditional IPA in maintaining character consistency were addressed by introducing low-level and region-level feature modeling techniques to ensure more coherent representations.
>
> b) The design of a transformer encoder adapter specifically accommodates these requirements, enhancing its adaptability within the DiT structure.
>
> Comparison with PhotoMaker: While PhotoMaker focuses on face customization, it does so with relatively simple facial features, without incorporating consistency mechanisms or accommodating multi-person scenarios. Our framework, on the other hand, is designed to handle more complex scenarios involving role customization with consideration for consistency across varied and intricate settings.

---

> ### Author Response · Authors · 2025-11-21
> **Weakness 3-5**
>
> - **Critically Incomplete Baseline Comparisons**
>
> **1. U-Net/SDXL-based SOTA methods like InstantID and PhotoMaker.**
>
> Firstly, InstantID and PhotoMaker are specialized methods focusing on face customization, heavily reliant on face ID embedding techniques. These methods, while state-of-the-art in their particular niche, are not designed for the broader scope of character customization. Character customization involves more complex and diverse features beyond facial characteristics, which is the primary focus of our work. Secondly, the domain of character customization has unique challenges and requirements that differ significantly from face customization. As indicated in our paper, the most advanced models addressing character customization are DiT-based. We have diligently compared our approach against these relevant DiT-based methods, such as ACE++, UNO, and EasyControl, which represent the current state-of-the-art within this specific domain.
>
>
> **2. DiT-native personalization methods (e.g., FLUX-Kontext)**
>
> As claimed in the first question, in our paper, we have compared several most-recent DiT-based sotas, and we provide the following analysis and comparison specifically regarding FLUX-Kontext:
>
> Firstly, we recognize that FLUX-Kontext is a contemporaneous development in DiT-native personalization methods. As such, we have conducted quantitative and qualitative comparative analyses between our approach and FLUX-Kontext (the qualitative results are updated in the appendix). While our method may not fully match FLUX-Kontext in overall performance, it exhibits clear advantages in both text controllability and inference speed. Moreover, our method achieves faster processing times, producing 1024-resolution images in just 18 seconds compared to 45 seconds for FLUX-Kontext. This efficiency stems from our lightweight adapter approach, which only adds computational load in the cross-attention section, in contrast to FLUX-Kontext's process of concatenating entire reference image sequences through all DiT layers.
>
> Secondly, we acknowledge that there are differences in comprehensive performance. It's important to note that FLUX-Kontext represents a larger, industrial-grade model that integrates image editing with customized data generation, benefiting from a higher scale and quality of data. Conversely, our data collection process was straightforward, with minimal filtering, making our approach more accessible and straightforward for widespread use. While FLUX-Kontext has not disclosed their data construction process, we look forward to their potential contribution to the community in terms of sharing insights into their data development, which could help push the field forward.
>
> | Method | IR↑ | CLIP-T↑ | Gemini-T↑ | CLIP-I↑ | DINO↑ | Gemini-I↑ | Time (s)↓ |
> | :--- | :---: | :---: | :---: | :---: | :---: | :---: | :---: |
> | Ours | 0.990 | **0.308** | **9.137** | 0.795 | 0.604 | 8.300 | **18** |
> | Kontext | **0.991** | 0.305 | 9.105 | **0.818** | **0.632** | **8.550** | 45 |
>
>
>
> - **Incomplete and Ambiguous Ablation Study on "dual-adapter" architecture**
>
> The design of the text adapter serves a crucial role in enhancing
> the separation of character features during the generation process by injecting visual features into
> textual features. It is well-known that diffusion models rely on text prompts to establish the initial
> layout of the image. By incorporating a text adapter, we aim to leverage this early influence on
> layout to effectively distinguish between different characters. In our ablation studies, we extracted
> attention masks between text tokens and image noise during the denoising process, as illustrated
> in Fig. S7. The absence of the text adapter resulted in the coupling of text tokens related to two
> different characters within the image space, causing the generated output to blend character features,
> thereby diminishing the consistency of each character. This observation is further supported by our
> quantitative experiments shown in Table S6, where the absence of the text adapter led to significant
> declines in consistency metrics such as CLIP-I, DINO, and Gemini-I.
>
> We have updated the analysis, table, and figure content in the appendix, in lines 665, 686, and 727.
>
> - **Unsatisfactory Qualitative Results & Model Bias: Despite claims of high fidelity, the qualitative results in the appendix (Figure S9) reveal significant failures in ID consistency**
>
> Thank you for the good suggestion. We acknowledge the observed issues, and particularly regarding changes in facial styles. We have noted these performance limitations in our appendix, specifically on line 697, under the limitations section. We recognize that maintaining style consistency, especially when dealing with small faces in reference images, is still a challenge in the area that requires further enhancement. Future work could focus on more specialized extraction of facial features to address these limitations and improve stylistic consistency.

---

> ### Author Response · Authors · 2025-11-21
> **Weakness 6**
>
> - **Dependence on Proprietary Data: The method's performance is inextricably linked to a massive, 10-million-sample proprietary dataset. This makes the results non-reproducible and makes it impossible to disentangle the contribution of the architecture from the contribution of the data.**
>
> Thank you for the question, we will address your concerns in the following aspects:
>
> Accessible data collection: Our dataset collection process is designed with minimal barriers, allowing for the use of any full-body photos of real or anime characters without extensive preprocessing, except for face detection. During the training of single-character InstantCharacter models, images with multiple faces were filtered out. The only straightforward filtering strategy enabled a substantial expansion of our dataset.
>
> Comprehensive framework: We have outlined the entire framework of our method, which includes image adapters for low-level and region-level feature consideration, text adapters for multi-character customization, and phased training strategies. These components are carefully designed to optimize character consistency, text controllability, and image fidelity, thus contributing significantly to the advancement of character customization research community.
>
> Dataset size ablation: We further conducted an ablation study during training stage 1 using various data sizes. In particular, we random split the all dataset to difference sizes. Results indicate that performance differences become negligible once the dataset reaches approximately 4-6 million samples, demonstrating the method's efficiency and suggesting that our results are not solely dependent on the size of the proprietary dataset. Moreover,we will committe to open-sourcing our code, enabling users to reproduce results with their own datasets. We also actively support the community in addressing any issues that arise, ensuring the reproducibility and adaptability of our research.
>
> | Data Size | IR↑ | CLIP-T↑  | Gemini-T↑ | CLIP-I↑ | DINO↑  | Gemini-I↑ |
> | :--- | :---: | :---: | :---: | :---: | :---: | :---: |
> | Stage1 (200w) | -1.152 | 0.231 | 4.125 | 0.885 | 0.832 | 9.050 |
> | Stage1 (400w) | -0.945 | 0.249 | 4.480 | 0.918 | 0.865 | 9.420 |
> | Stage1 (600w) | -0.885 | 0.256 | 4.605 | **0.928** | **0.879** | 9.550 |
> | Stage1 (800w) | -0.874 | 0.252 | **4.631** | 0.926 | 0.873 | 9.575 |
> | Stage1 (1000w)| **-0.870** | **0.258** | 4.622 | **0.928** | 0.877| **9.586** |

---

### Author Response · Authors · 2025-11-23

We extend our gratitude to all the reviewers for their valuable insights and contributions, which have significantly enhanced the quality of our paper. We have carefully addressed all your concerns and updated the manuscript accordingly. Should you have any further questions or wish to discuss any aspect of our work, please feel free to reach out. We welcome and look forward to continued dialogue.

---

### Meta-Review · Area_Chair_penm · 2025-12-28

**Summary:**

The reviewers' concerns are mainly about 1) novelty, 2) methodology, 3) evaluation

1. Reviewers 3n5x and 8u1R think that the proposed method is a combination of existing techniques (e.g., IP-Adapter's cross-attention injection, PhotoMaker's fused embeddings per reviewer 3n5x), hence having limited novelty. Reviewer 3n5x also mention that the paper's claim of being the "first DiT-based framework" is not appropriate.

2. Reviewers flag that ablation studies are insufficient to demonstrate the effectiveness of the components, and the reliance of a massive non-public dataset.

3. Reviewers point out that comparisons to SOTA methods (e.g., InstantID) are missing, and the proposed method does not show significant improvement over existing works.

**Reviewer Concerns:**

The authors resolve the reviewers' concerns by providing comparison with additional baselines, ablation studies on the text adapter, and scalability studies. However, the concerns related to novelty, reproducibility, and method effectiveness are not fully resolved.

**Reviewer Scores:**

This paper receives initial ratings of (4, 4, 4, 6). Since concerns are partially resolved and no reviewers explicitly mention that score would be increased, the AC anticipates a final score of (4, 5, 5, 6).

---

### Decision · Program_Chairs · 2026-01-26

Reject